# Hazard source detection of longitudinal tearing of conveyor belt based on deep learning

**Yimin Wang**[1,2,3], **Changyun Miao**[1,3]*, **Di Miao**[4], **Dengjie Yang**[1], **Yao Zheng**[1]

**1** School of Mechanical Engineering, Tiangong University, Tianjin, China, **2** Tianjin Electronic Information College, Tianjin, China, **3** Tianjin Photoelectric Detection Technology and System Key Laboratory, Tiangong University, Tianjin, China, **4** School of Electronic Engineering, Tianjin University of Technology and Education, Tianjin, China

* miaochangyun@tiangong.edu.cn

**Data Availability Statement:** Data are available at https://aistudio.baidu.com/aistudio/datasetdetail/187806 Author-generated code is available at https://gitee.com/rugeweiwu/ssdshufflenetv2.

## Abstract

Belt tearing is the main safety accident of belt conveyor. The main cause of tearing is the doped bolt and steel in the conveying belt. In this paper, the bolt and steel are identified as the Hazard source of tear. In this paper, bolt and steel are defined as the risk sources of tearing. Effective detection of the source of danger can effectively prevent the occurrence of conveyor belt tearing accidents. Here we use deep learning to detect the hazard source image. We improved on the SSD(Single Shot MultiBox Detector) model. Replace the original backbone network with an improved Shufflenet_V2, and replace the original position loss function with the CIoU loss function. Additionally, it compares this new approach to previous methods. The proposed model has surpassed other state-of-art methods with more than 94% accuracy. In addition, when deployed without GPU acceleration, the detection speed can reach 20fps. It can meet the requirements of real-time detection. The experimental results show that the proposed model can realize the online detection of hazard sources, so as to prevent longitudinal tearing of conveyor belt.

## Introduction

Conveyor belt tearing accident is one of the most frequent safety accidents in coal mine.If there is a tear accident, usually accompanied by equipment damage, resulting in production and casualties. According to research, the primary cause of the conveyor belt's longitudinal tearing is damage to the belt by foreign objects introduced during material transportation, such as gangue, scrap steel, bolt, or wood. Therefore, the aforementioned foreign objects can be considered a risk factor for the conveyor belt to tear longitudinally and must be found.

Currently, manual detection, metal detector detection, radiographic detection [1], spectral detection, and other approaches are the main ways to find conveyor belt hazards that cause longitudinal ripping. Manual detection techniques take much time, are ineffective, and could be dangerous. Metal detectors have a complicated detection system. Generally, iron metal detectors are utilized, which can only detect foreign objects with high iron content and cannot

**Funding:** DM, Planning Projects of Science and Technology Support of Tianjin, No. 17YFZCSF01210. Funders provide financial support for this research.

**Competing interests:** The authors have declared that no competing interests exist.

detect many danger sources. Zhao et al. [2] used X-rays to differentiate and find alien substances other than coal. This approach can discern between distinct item types using a differential energy absorption coefficient brought on by the different elements present in various substances. However, there are issues with expensive equipment and radiation that are detrimental to human health. Song et al. [3] proposed a technique to discriminate between coal and gangue incorporating near-infrared and far-infrared analyses. This technique can efficiently discriminate between coal and gangue, but it has no discernible impact on identifying other sources of hazards. It is challenging for the techniques above to be widely applied in real-world application scenarios because each has its drawbacks.

Target identification technology based on machine vision has advanced considerably with the advancement of machine vision technology [4–8]. Ali et al. [9] used a two-layer framework of the GoogLeNet and YOLOv3 models for tumor detection in MRI. Khan et al. [10] applied the VGG-NIN model to the analysis of diabetic retinopathy. Vulli et al. [11] used DenseNet-169 model to accurately predict lymph node metastasis of breast cancer.

Machine vision detection has also simultaneously evolved into the main belt conveyor safety monitoring study's central area. A conveyor belt image processing algorithm was proposed by Yang et al. [12] which uses an LED light source to illuminate the lower surface of the conveyor belt. A linear CCD camera to capture images of the lower surface of the conveyor belt and a conveyor belt image processing algorithm to detect longitudinal tearing and other faults. Gao et al. [13] used an intelligent inspection robot video monitoring belt foreign item automatic recognition technique. Its fundamental idea is to use RGB transformation and foreground and background matrix calculation to compare the foreign target object to the background. The aim is the difference that is greater than the threshold. Su et al. [14] proposed an improved LeNet-5 model for image classification of coal and coal gangue. Compared with traditional methods, this model improves the recognition accuracy. Li et al. [15] proposed a method for detecting coal and gangue in images that uses the YOLOv3 algorithm to identify coal gangues. Xiao et al. [16] suggested using a more objective approach to image segmentation based on the RDU net model, which combines the residual structure of the convolutional neural network with the dunet model. Zhang et al. [17] presented a method for detecting foreign objects in coal using machine vision based on an attention neural network. By utilizing visualization technology, this technique developed a CNN with an attention module to successfully detect foreign objects in coal. The detection speed increased to 15fps with GPU acceleration. However, the model was too huge, and the detection was too GPU-dependent. A multimodal imaging-based conveyor belt foreign object detection method has been proposed by GURAV Saran [18] et al. It collects images using a polarizing camera and identifies foreign objects by displaying different colours for various materials underneath the camera. However, the system recognition delay is 1 second, and the real-time performance is subpar. The challenge with current machine learning-based detection techniques is that accuracy and real-time are frequently incompatible. Great hardware is needed to run the complex model with high detection accuracy. Real-time detection cannot be done without its deployment in a GPU environment. Therefore, it is essential to research a technique that can accurately and quickly identify the conveyor belt's risk of longitudinal ripping. The main contributions of this study are as follows;

1. To build a model that can accurately identify the risk source of conveyor belt tearing is established.

2. Fine-tuning the Shufflenet_V2 network to reduce the number of parameters on the network.

3. The SSD model was improved with the improved Shufflenet_V2 network and the CIoU function to achieve higher accuracy. The model hardware requirements are reduced so that it can be deployed on platforms without GPU.

The entire manuscript is further divided into the following sections:The Materials and methods used in the current study are presented in Section 2. The observations and the results are discussed in Section 3. And finally, the Conclusion and Future Scope are presented in Section 4.

## Materials and methods

### SSD (single shot multibox detector) model

Target detection flow methods now come in two varieties: one stage and two stages. The target detection process is finished chiefly using a complete convolutional neural network, distinguished by high accuracy in the two-stage method, represented by fast R-CNN [19]. Its shortcomings include a high percentage of false alarms, a lengthy training period, and poor detection speed. The one-stage method samples the image using frames of various scales at various locations and then utilizes CNN to extract features for recognition and classification. It is distinguished by its fast speed and can detect longitudinal tear hazards in real-time. A typical one-stage application algorithm is the SSD [20] model. Although the low detection accuracy of small-size targets is the principal drawback of the SSD algorithm, most of the hazard sources for conveyor belt longitudinal ripping are large, sharp objects. Therefore this issue has little practical significance. SSD model is also well applied in longitudinal tearing detection of conveyor belts [21]. The SSD algorithm uses an anchor mechanism, eliminating a candidate box. Every point in the candidate region becomes the central point because of the anchor mechanism. The final two FC layers of VGG16 are changed into conv6 and conv7 layers, which are employed as the backbone network for feature extraction. Later, add 6 convolutional layers. For classification, SSD takes six feature maps from the previous convolution layer and concatenates all results. Finally, output the feature map using the global average pool. Fig 1 depicts the network structure of the SSD model.

The SSD loss function adopts the multi-loss fusion method: the weighted sum of SSD localization loss (loc) and confidence loss (conf). And the position loss function is

$$L_{loc}(p, b) = \sum_{i \in \{x,y,w,h\}} smooth_{L1}(p_i - b_i) \tag{1}$$

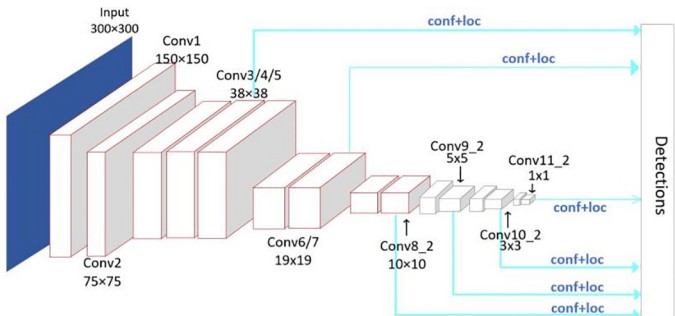

**Fig 1. Network structure of SSD model.**

Where $smooth_{L1}$ is

$$smooth_{L1}(x) = \begin{cases} 0.5x^2 & |x| < 1 \\ |x| - 0.5 & other \end{cases} \tag{2}$$

$b$ represents the real box coordinates, $b = (b_x, b_y, b_w, b_h)$

$p$ represents the predicted box coordinates, $p = (p_x, p_y, p_w, p_h)$

In other words, calculate the losses for the four parameters x, y, w, and h, and then add those losses as position loss functions. This disregards the overlap between the genuine and predicted boxes and the correlation between the four parameters.

The SSD model is enhanced as follows in order to increase the precision and speed of the identification of the conveyor belt's longitudinal ripping hazard source.

1. The original VGG16 network is replaced with the upgraded Shufflenet_V2 network, which significantly simplifies the model's calculation and enhances the system's real-time performance while preserving accuracy.

2. To increase the precision of model detection, the original position loss function is replaced with the CIoU loss function.

3. Substitute 3x3dwconv for all 3x3convs of the SSD model, excluding the backbone network.

## Improved Shufflenet_V2 network

The fundamental idea behind Shufflenet is to shuffle several channels to overcome the drawbacks of group revolution. The model's accuracy is maintained while being considerably reduced through the use of two procedures, pointwise group revolution and channel shuffle [22], and its structure is depicted in Fig 2.

Stage modules make up the majority of the Shufflenet V2 network structure. These modules are primarily based on a DownSampling block and a normal block [23]. The channel split operation splits the conventional normal block into two channels. The concatenation procedure is used for integration after the data from the two channels have been calculated separately. The

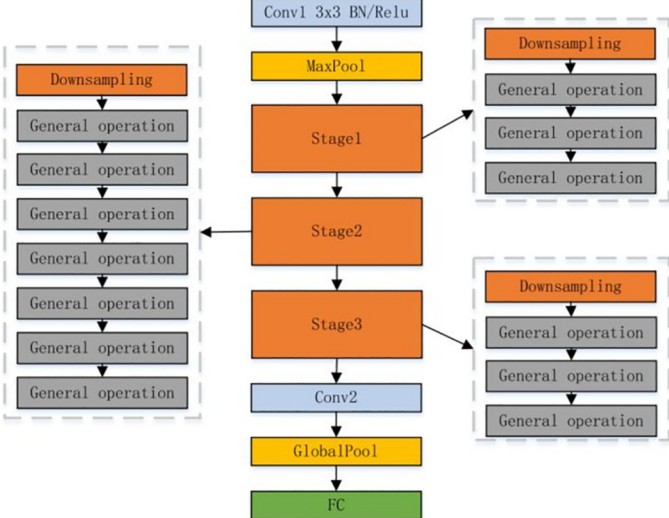

**Fig 2. Shufflenet_V2 network structure.**

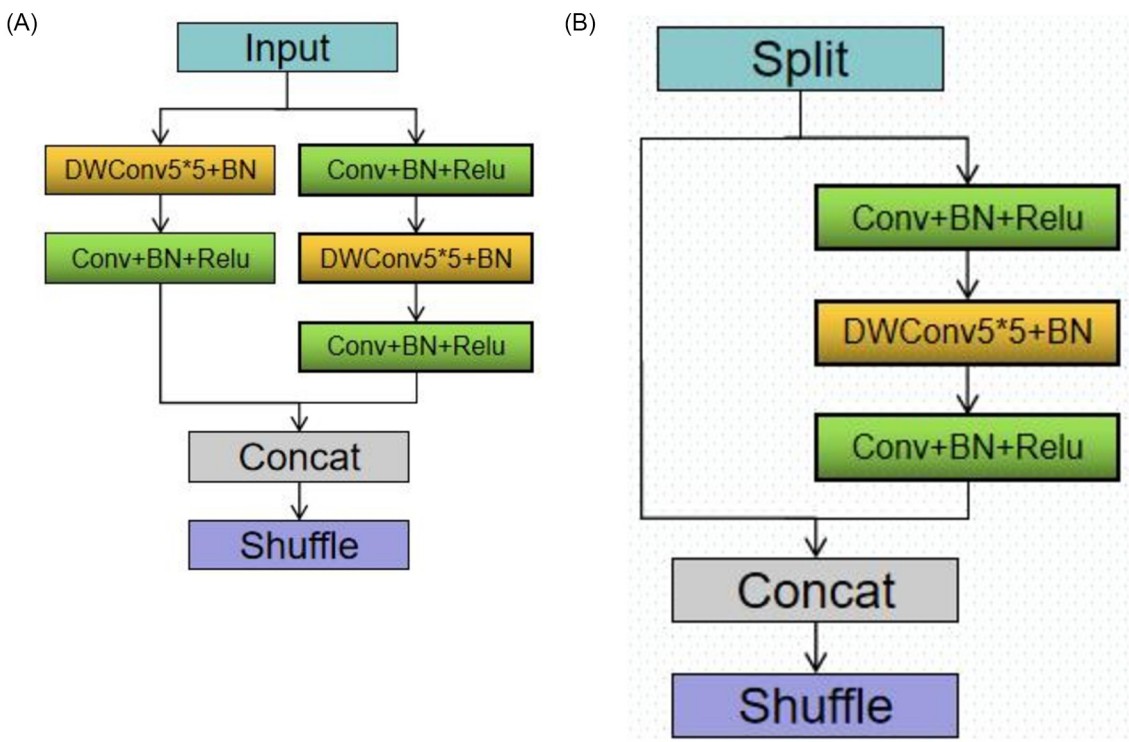

**Fig 3. Shufflenet_V2 network main block (a) DownSampling block (b) Normal block.**

order is finally disturbed to ensure that each channel goes through the bottleneck structure once after the subsequent channel shuffle. The DownSampling block, which replaces the pooling procedure, doubles the number of picture channels while maintaining the same number of input and output channels and feature map size.

There are two ways to optimize and enhance the model: one is to increase the model's accuracy, and the other is to shorten the model's runtime. The fundamental technique of increasing the convolution kernel has the potential to immediately increase the model's accuracy. We can see that the calculation of DepthWise convolution (DWConv) is small. In order to expand the DWconv convolution kernel and replace the original 3*3 convolution kernel with a 5*5 convolution kernel, we decided to analyze the calculation of the Shufflenet_V2 network. In Fig 3, the improved structure is displayed.

Fig 3(a) shows the DownSampling block. On the left side, we have a DWConv operation of 5*5, stride size 2, and then BatchNorm(BN). Finally by 1x1 convolution+BN+ReLU; The right-hand half is 1x1 convolution+BN+ReLU. And then use a DWConv of 5*5, the stride size is 1, and then after BN. Finally another 1x1 convolution+BN+ReLU; After finishing the two-side operation, Concat and channel shuffle the result. Fig 3(b) shows the normal block. First it goes through a channel split. The right half goes through 1x1 convolution+BN+ReLU. After a DWConv5*5+BN. Finally another 1x1 convolution+BN+ReLU. After all the operation is the Concat operation and the channel shuffle.

Customizing the model is the standard method of decreasing the model's runtime. Stage modules make up the Shufflenet_V2 network's core. The stage module uses a combination structure that consists of a DownSampling block superimposed multiple times over a standard operation block. Fig 2 depicts its structure. To speed up the model operation, the model is

**Table 1. The architecture of improved Shufflenet_v2 network.**

| Layer | | Kernel Size | Stride | Tensor Size |
|---|---|---|---|---|
| Conv | | 3×3 | 2 | 150×150 |
| MaxPool | | 3×3 | 2 | 75×75 |
| Stage1 | Downsampling | | 2 | 38×38 |
| | Normal*3 | | 1 | 38×38 |
| Stage2 | Downsampling | | 2 | 19×19 |
| | Normal*3 | | 1 | 19×19 |
| Stage3 | Downsampling | | 2 | 10×10 |
| | Normal*3 | | 1 | 10×10 |
| Conv | | | 1 | 10×10 |
| GlobalPool | | | | 1×1 |
| FC softmax | | 1000D | | |

trimmed by lowering the standard operation block of stage 2's repeat times from 7 to 3 times. The details of the improved Shufflenet_V2 are depicted in Table 1.

## CIoU loss function

Zheng et al. [24] proposed the CIoU loss function in 2020. Fig 4 depicts the placement diagram for both the actual Box B and the BGT prediction box.

Considering the overlapping area, center point distance, and aspect ratio of the real box and the prediction box, the CIoU position loss function substantially increases the convergence speed and accuracy. As a result, the CIoU loss function consists of three components:

The loss of the overlapping area between the real box and the prediction box makes up the first component

$$L_1 = 1 - IoU \tag{3}$$

The intersection and union ratio of the real box and the prediction box are represented by the IOU function, which has the following formula and is the most often used position loss

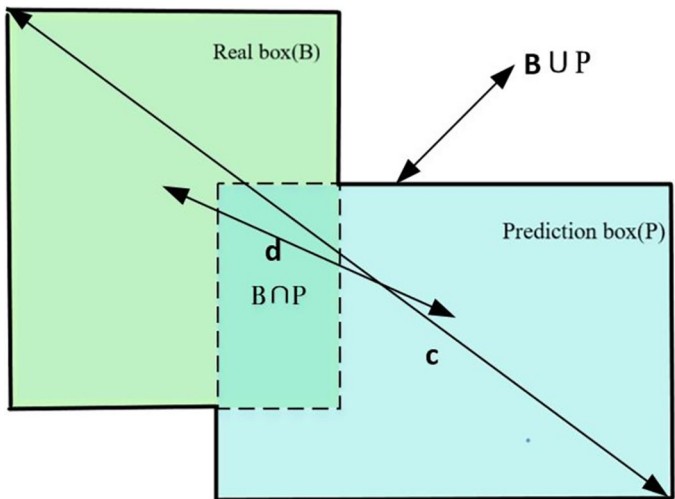

**Fig 4. Schematic diagram of the position of real Box B and prediction box BGT.**

function in target detection:

$$IoU = \frac{|B \cap P|}{|B \cup P|} \tag{4}$$

The loss in normalized distance between the center point of the prediction box and the actual box makes up the second component

$$L_2 = \frac{d^2}{c^2} \tag{5}$$

Where C denotes the diagonal distance between the smallest outer rectangle of the prediction box and the real box, and d denotes the distance between the center point of the real box and the prediction box. The third component is the aspect ratio loss between the forecast and real box.

$$L_3 = av \tag{6}$$

V measures the similarity of the aspect ratio, which is determined by using the weight function A.

$$v = \frac{4}{\pi^2} \left( \arctan \frac{w_p}{h_p} - \arctan \frac{w_b}{h_b} \right)^2 \tag{7}$$

$$a = \frac{v}{(1 - IoU) + v} \tag{8}$$

$w_b$ and $h_b$ stand for the width and height of the prediction box, respectively, while $w_p$ and $h_p$ Represent the width and height of the real box.

The aforementioned three elements together form the full definition of the CIoU loss function:

$$Loss(CIoU) = 1 - \frac{|B \cap P|}{|B \cup P|} + \frac{d^2}{c^2} + av \tag{9}$$

5 Improved SSD_Shufflenet_V2 model

Design improved SSD Shufflenet_V2 model, which replaces the backbone network VGG16, conv6 and conv7 of SSD model, using improved Shufflenet_V2 network. SSD Shufflenet_V2 uses an improved Shufflenet_V2 model part from conv1 to conv14, removing Shufflenet_V2 final global maximum pooling and FC. After the improved Shufflenet_V2 network adds conv15 1, Conv15 2... Fig 5 depicts its structure.

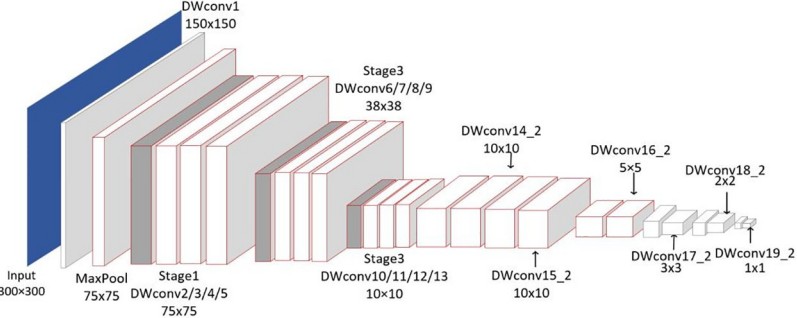

**Fig 5. Improved SSD_Shufflenet_V2 network structure.**

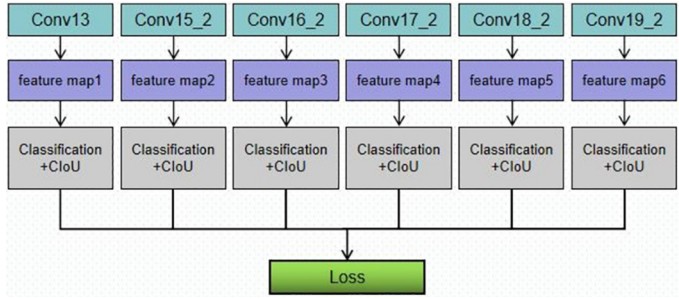

**Fig 6. Loss calculation.**

The 6-layer feature map used in extract Conv13, Conv15 2, and Conv19 2 is utilized to calculate the classification loss and location loss before calculating the final loss function. As depicted in Fig 6.

## Results and discussion

### Experimental platform

Our team has constructed a complete set of speed-adjustable belt conveyor trial equipment, which includes a system for circulating material made up of two belts that are 25 meters long and two that are 5 meters long, as well as several belt conveyor safety protection systems. The system is equipped with the following experimental devices:Conveyor belt dating detection device based on X-ray; Conveyor belt deviation detection equipment; Material flow detection device; Belt speed detection control device; Inspection robots and robotic arms and other equipment. Fig 7 shows a picture of part of the experimental platform. An industrial camera is positioned directly above the conveyor belt to monitor the material quality in real-time and identify potential hazard sources.

RTX3090 GPU and Ubuntu 18 operating systems serve as the model training environment. The Python programming language was used to develop the model in this paper, and the deep learning technology discussed in the present study was constructed based on the Caffe deep learning framework. Windows 10 operating system, an Intel i9-12900K@3.19 GHz CPU, 32G of memory, and a lack of a GPU environment make up the model test environment. The training hyperparameters adopted are shown in Table 2.

### Data pre-processing and augmentation

One thousand sample photographs with coal in the background, including gangue, bolt, wood, and steel, were collected using industrial cameras. The sample is expanded by image

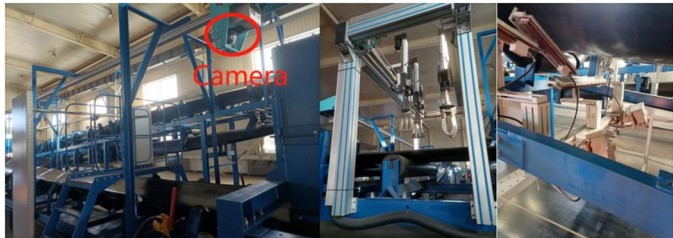

**Fig 7. Belt conveyor experimental platform.**

**Table 2. The training HyperParameters adopted.**

| Parameters | Value |
|---|---|
| Initial learning rate | 0.01 |
| Momentum | 0.9 |
| Batch | 32 |
| Epoch | 300 |
| Input size | 300×300 |

enhancement, horizontal flip, rotation and cropping. Generate a data set of 10,000 photos, including 2400 gangue images, 2800 anchor bolt images, 2400 wood images, and 2400 waste steel images. LabelImg software was used to annotate the images, as shown in Fig 8. 1000 images are chosen as the test data set, while 9000 images are chosen as the training data set.

## Evaluation metrics

In order to better compare the performance of models. The main performance indicators adopted are "Precision" and "Accuracy". First, we need to describe TP (True Positive), TN (True Negative), FP(False Positive), FN(False Negative). TP is the number of correctly categorized records belonging to the positive class, while TN is the number of correctly categorized records belonging to the negative class. FP and FN represent the number of incorrectly categorized records belonging to the positive and the negative classes, respectively.

$$\text{Precision} = \text{TP}/(\text{TP} + \text{FP}) \tag{10}$$

$$\text{Accuracy} = (\text{TP} + \text{TN})/(\text{TP} + \text{TN} + \text{FP} + \text{FN}) \tag{11}$$

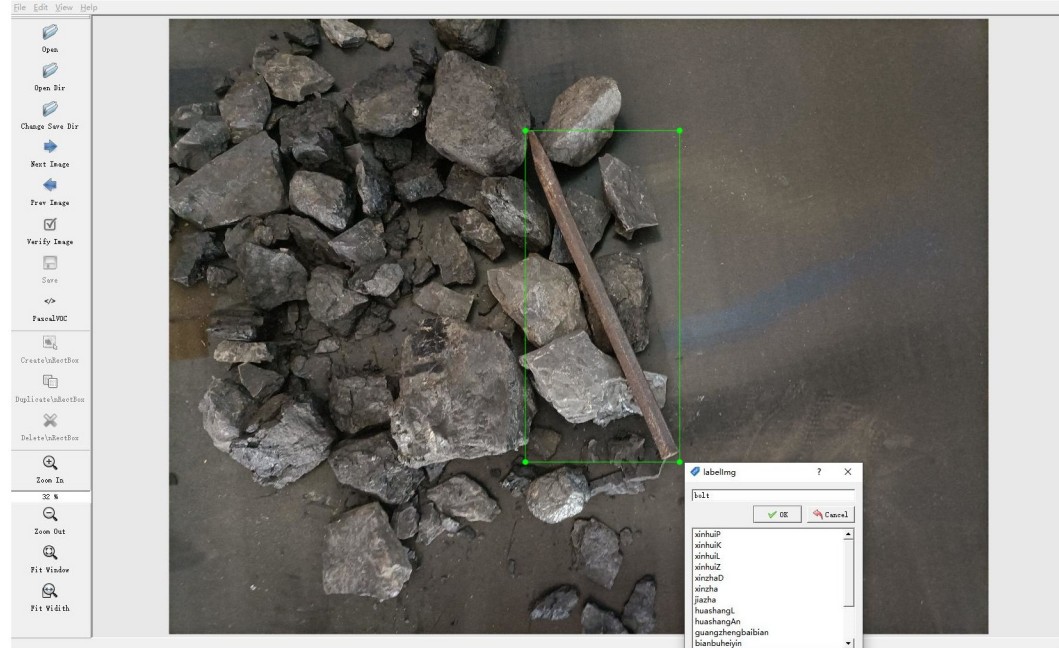

**Fig 8. LablImg software annotates images.**

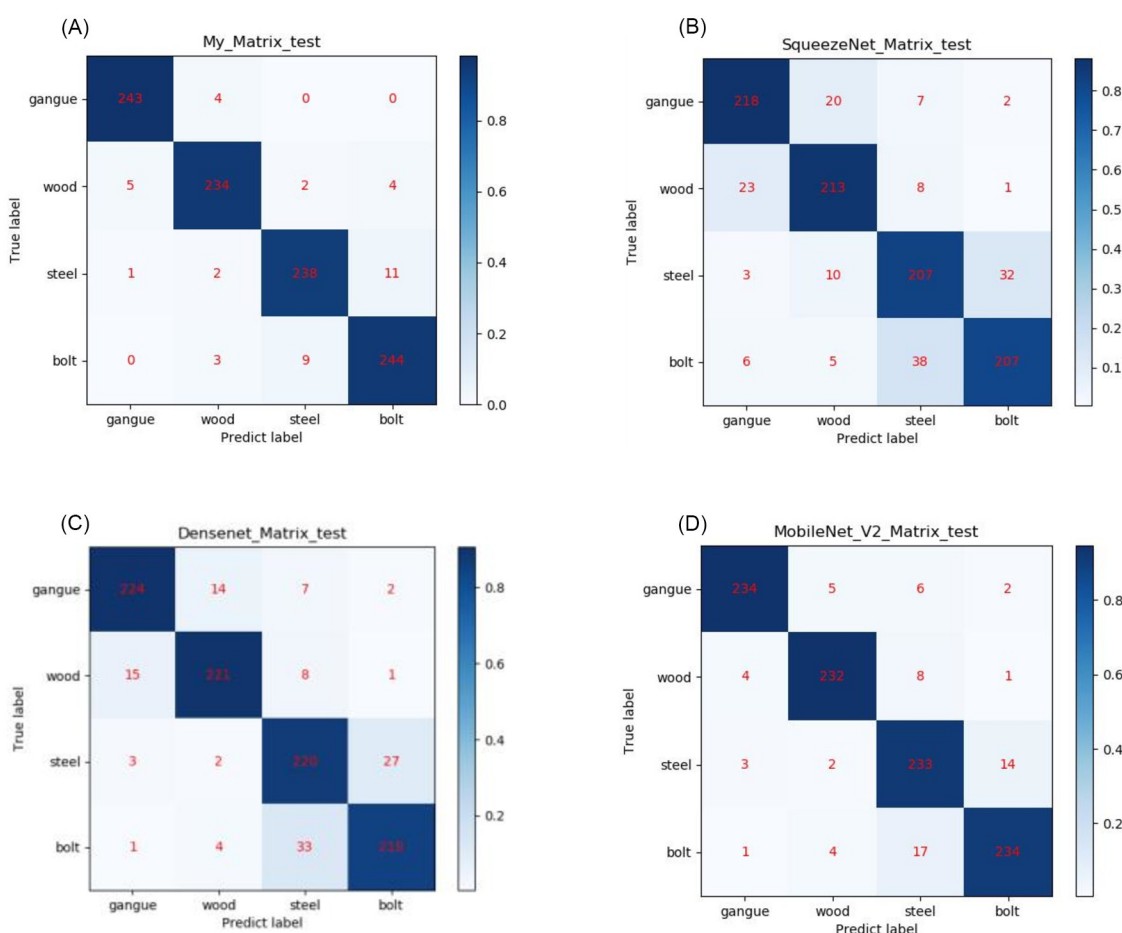

**Fig 9. The confusion matrices: (a) the models of enhanced Shufflenet_v2; (b) SqueezeNet model; (c)Densenet model; (d) MobileNet_V2 model.**

**Verify the benefits of the improved Shufflenet_v2 network.** Squeezenet, Mobilenet, and Densenet are a few examples of lightweight convolutional neural networks that are compared with the enhanced Shufflenet V2 network. Obtain the confusion matrix of the identification outcomes for four types of hazard sources: bolt, gangue, wood, and steel, depicted in Fig 9.

The graph shows the test results of 1000 samples. Taking Fig 9(a) as an example. An output is called TP(gangue) when the model recognizes the instance as gangue, and the actual output is gangue. An output is called TN(gangue) when the model recognizes the instance as gangue, and the actual output is not gangue. Then TP(gangue) = 243, FP(gangue) = 6. Calculated Precision(gangue) = 97.6%. Also, Precision(wood) = 96.3%, Precision(steel) = 95.6%, and Precision(bolt) = 94.2%. The final Accuracy is 95.9%. The comparison between the model in this paper and other models is shown in Table 3.

Table 3 shows that the enhanced Shufflenet_V2 network, which has a greater accuracy rate of 95.9% than other comparison networks, is utilized to ensure the accuracy rate under the assumption that its parameters and computational complexity are lowered.

**Verify the advantages of the CIoU loss function.** Improved SSD_Shufflenet_V2 utilizing CIoU as the position loss function in experiment 2. Fig 9 displays the target detection results. The model is different from the conventional SSD model, and SSD_Shufflenet_V2 does not

**Table 3. Comparison and verification of various lightweight convolutional neural networks.**

| Model | Loss | | Accuracy | |
|---|---|---|---|---|
| | Training | Test | Training | Test |
| SqueezeNet V1.1 | 0.1856 | 0.3238 | 90.2% | 84.5% |
| MobileNet V2(0.75) | 0.0763 | 0.1589 | 98.1% | 93.3% |
| Densenet(40) | 0.1108 | 0.2188 | 94.2% | 88.3% |
| Enhanced Shufflenet_v2 | 0.0652 | 0.1242 | 98.9% | 95.9% |

use CIoU as the location loss function model (from now on referred to as CIoU model), is used to analyze and verify the effect of target detection.

Fig 10(a) depicts the object detection diagram for the model used in this study; (b) depicts the object detection diagram without the use of the CIoU model; and Fig 10(c) depicts the object detection diagram for the conventional SSD model. When Fig 10(b) and 10(c) are compared, it can be seen that there is no discernible difference between the detection accuracy of the SSD_Shufflenet_V2 model created by Shufflenet_V2 and VGG16. When Fig 10(a), 10(b) and 10(c) are compared, the model in this paper compares the unused CIoU model and the traditional SSD model, which makes the selected target area more accurate without affecting the detection accuracy. Fig 11 depicts the network detection accuracy curve used in this study. The average detection accuracy of this paper is 94.6%

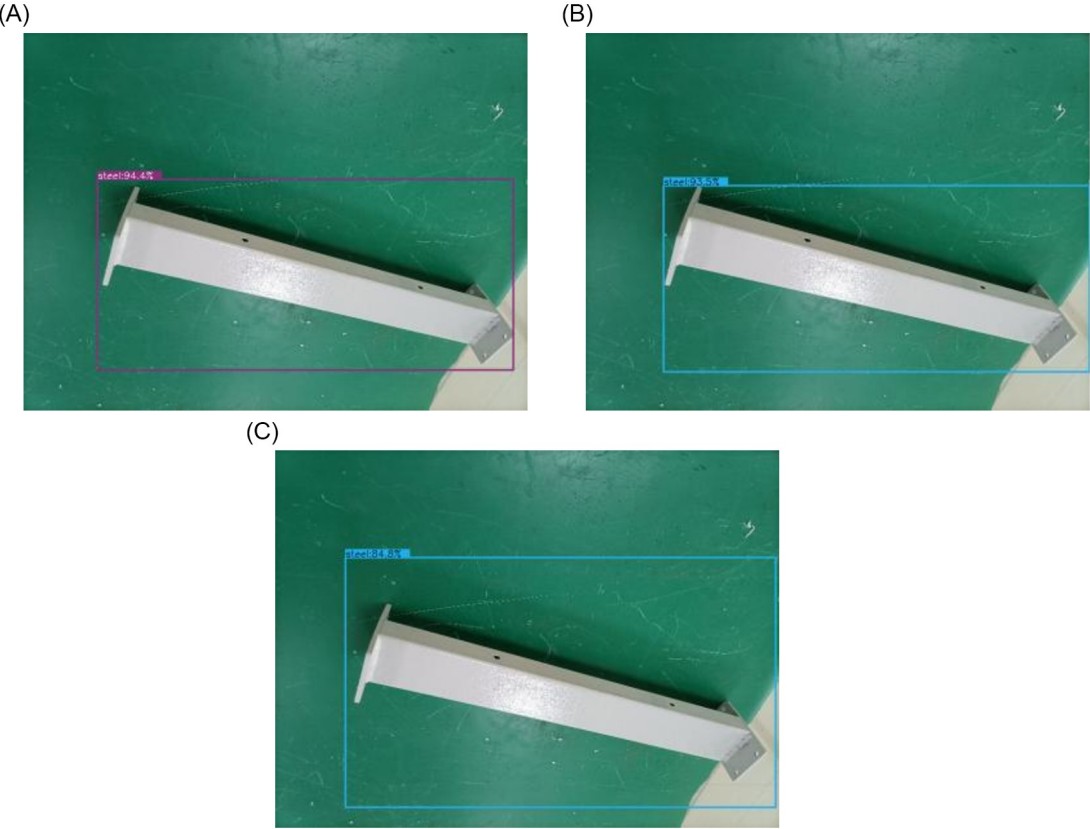

**Fig 10. The object detection diagram: (a) the models of this paper; (b) without the CIoU model; (c) SSD model.**

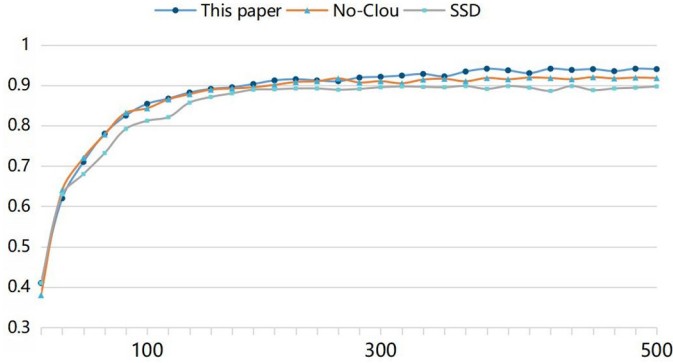

**Fig 11. Accuracy rate curve of the method proposed in this paper.**

**Performance comparison with other ensemble methods.** We compared our method with four other object detection methods. For example, convolutional neural network model [18], RDU-Net model [17], YoloV3 model and improved LeNet method [14]. The results are shown in Table 4. Based on the comparison of the proposed model with previous methods, it can be concluded that our method is superior to the methods listed below.

The model presented in this study is applied to the experimental platform used by the research team. The video target recognition verification experiment is carried out on-site using a belt conveyor while using the opencv+python environment to call the camera. The outcomes are displayed in Fig 12. The findings demonstrate that this technique is effective at achieving belt conveyor hazard identification while being video monitored, and the detection speed can exceed 20fps.

## Conclusion

This work aimed to solve the problem of detecting the hazard source of longitudinal tearing of conveyor belt. In this paper, we proposed a upgraded SSD (single shot multibox detector) model, which uses an improved original backbone network is replaced with Shufflenet_V2. At the same time, we use CIoU loss function to replace the original position loss function. Use improved models to achieve greater accuracy and time efficiency. The proposed method can accurately filter the data, extract the features, and identify the source of danger in the image. The accuracy rate is more than 94%. The detection speed can exceed 20fps when used in an environment without GPU acceleration. It is concluded that this method can meet the requirements of online warning detection of longitudinal tear. In future studies, we will study the image quality problems caused by the environment. Add the analysis of image deblurring

**Table 4. Performance comparison between other methods and our model.**

| Model | Precision | | | | Accuracy |
|---|---|---|---|---|---|
| | gangue | wood | steel | bolt | |
| CNN [18] | 93.6% | 94.8% | 92.6% | 92.5% | 93.4% |
| RDU-Net [17] | 92.8% | 93.5% | 91.5% | 92.1% | 92.5% |
| YoloV3 | 90.1% | 91.2% | 86.8% | 90.4% | 89.6% |
| LeNet [14] | 90.4% | 91.8% | 89.8% | 91.2% | 90.8% |
| This Model | 94.8% | 96.5% | 93.1% | 94.1% | 94.6% |

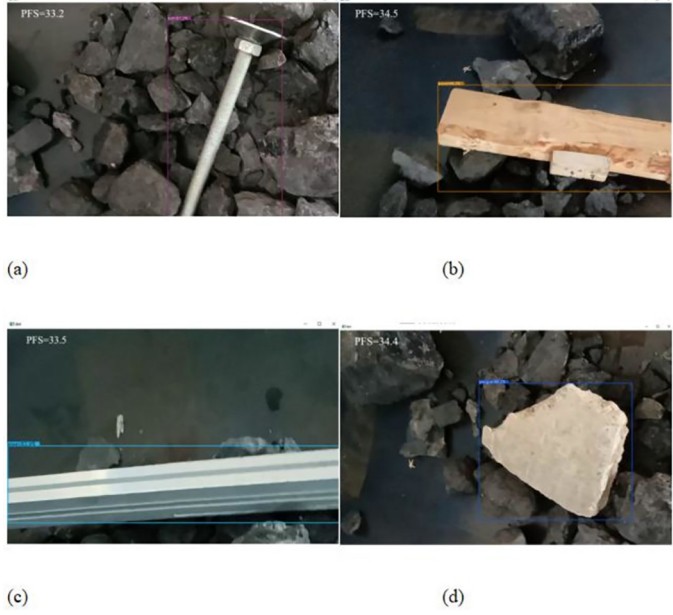

**Fig 12. Video target recognition verification of this model.**

method. The method in this paper will be applied to other scenes of conveyor belt detection to further verify the superiority of the model.

## Author Contributions

**Conceptualization:** Yimin Wang.

**Data curation:** Di Miao.

**Formal analysis:** Yao Zheng.

**Funding acquisition:** Yao Zheng.

**Methodology:** Dengjie Yang.

**Project administration:** Changyun Miao.

**Software:** Yimin Wang.

**Validation:** Di Miao.

**Visualization:** Dengjie Yang.

**Writing – original draft:** Yimin Wang.

**Writing – review & editing:** Changyun Miao.

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
