## [Decision Letter · Decision Letter 0]

15 Dec 2022

PONE-D-22-30806Hazard source detection of longitudinal tearing of conveyor belt based on deep learningPLOS ONE

Dear Dr. Wang,

Thank you for submitting your manuscript to PLOS ONE. After careful consideration, we feel that it has merit but does not fully meet PLOS ONE’s publication criteria as it currently stands. Therefore, we invite you to submit a revised version of the manuscript that addresses the points raised during the review process.

We look forward to receiving your revised manuscript.

Kind regards,

Muhammad Fazal Ijaz

Academic Editor

PLOS ONE

Journal Requirements:

"DM,Planning Projects of Science and Technology Support of Tianjin,No. 17YFZCSF01210"

Reviewers' comments:

Reviewer's Responses to Questions

**Comments to the Author**

1. Is the manuscript technically sound, and do the data support the conclusions?

Reviewer #1: Partly

Reviewer #2: Yes

2. Has the statistical analysis been performed appropriately and rigorously? 

Reviewer #1: Yes

Reviewer #2: Yes

3. Have the authors made all data underlying the findings in their manuscript fully available?

Reviewer #1: Yes

Reviewer #2: Yes

4. Is the manuscript presented in an intelligible fashion and written in standard English?

Reviewer #1: Yes

Reviewer #2: Yes

5. Review Comments to the Author

Reviewer #1: The overall impression of the technical contribution of the current study is reasonable. However, the Authors may consider doing necessary amendments to the manuscript for better comprehensibility of the study.

1. The abstract must be re-written focusing on the technical aspects of the proposed model, the main experimental results, and the metrics used in the evaluation. Briefly discuss how the proposed model is superior.

2. The contribution of the current study must be briefly discussed as bullet points in the introduction. And motivation must also be discussed in the manuscript.

3. The overall organization of the manuscript is not discussed anywhere in the manuscript. Please add the same in the introduction section of the manuscript.

4. Introduction section must discuss the technical gaps associated with the current problem.

5. The literature section is missing, authors are recommended to incorporate the same for better comprehensibility of the study.

6. Authors may include some of the relevant studies on deep learning like https://doi.org/10.3390/bios12060393 and https://doi.org/10.3390/s22082988

7. Authors may provide the architecture/block diagram of the proposed model for better comprehensibility of the proposed model concerning to various aspects of the proposed model.

8. More explanation of the proposed model is desired on technical grounds.

9. What is the size of the input image that is considered for processing and the size of the kernels?

10. The important details like the size of the input/tensor/kernel must be discussed, and whether authors have used Stride 1 or Stride 2 must be presented. What type of activation function is being used in the current study.

11. For how many epochs does the proposed model executed. what is the initial learning rate and after how many epochs does the model's learning rate saturated.

12. Authors may present the loss functions for better comprehensibility of each of the models used in the proposed model.

13. Majority of the figures lack the clarity, they quality is fair but they must be explained in the text and the figures must be cited. Where is the graph for testing loss and accuracy presented in the study.

14. Please discuss more on the implementation platform and the dataset details as two sub-sections in the manuscript.

15. What are the cases assumed as TP, TN, FP, FN (confusion matrix) in the current study.

16. Authors must provide the details of hyper parameters like training loss, testing loss, training accuracy and testing accuracy.

17. More comparative analysis with state-of-art models is desired.

18. By considering the current form of the conclusion section, it is hard to understand by PLOS ONE Journal readers. It should be extended with new sentences about the necessity and contributions of the study by considering the authors' opinions about the experimental results derived from some other well-known objective evaluation values if it is possible.

Reviewer #2: In this paper, authors presented a detection method of longitudinal tearing hazard sources of belt conveyor based on deep learning in order to address the issues of poor accuracy and real-time in the identification of longitudinal tearing hazard sources of belt conveyor. However, there are some limitations that must be addressed as follows.

1. Both the abstract and introduction are not professionally written. Some sentences in abstract should be modified to make it more attractive for readers

2. In Introduction section, it is difficult to understand the novelty of the presented research work. This section should be modified carefully. In addition, the main contribution should be presented in the form of bullets.

3. The authors should also discuss the following works, which are about deep neural networks and image classification: ‘A Two-Tier Framework Based on GoogLeNet and YOLOv3 Models for Tumor Detection in MRI’, ‘Diabetic Retinopathy Detection Using VGG-NIN a Deep Learning Architecture’, ‘Traffic accident detection and condition analysis based on social networking data’, ‘Classification of skin disease using deep learning neural networks with MobileNet V2 and LSTM’

4. Figures are blurred, it is difficult to read these figures, their quality should be improved (see fig 3)

5. Captions of the Figures not self-explanatory. The caption of figures should be self-explanatory, and clearly explaining the figure. Extend the description of the mentioned figures to make them self-explanatory.

6. The conclusion section should be revised. In addition, the future work should be properly discussed.

7. The whole manuscript should be thoroughly revised in order to improve its English.

6. PLOS authors have the option to publish the peer review history of their article (what does this mean?). If published, this will include your full peer review and any attached files.

Reviewer #1: No

Reviewer #2: No

---

## [Author Response · Author response to Decision Letter 0]

13 Feb 2023

Reviewer#1, Concern # 1: 

Author response: The abstract must be re-written focusing on the technical aspects of the proposed model, the main experimental results, and the metrics used in the evaluation. Briefly discuss how the proposed model is superior.

Author action: We updated the manuscript by modify the abstract . 

Reviewer#1, Concern # 2: 

Author response: The contribution of the current study must be briefly discussed as bullet points in the introduction. And motivation must also be discussed in the manuscript.

Author action: We updated the manuscript by the contribution. We have added the contribution of this study and other changes.

Reviewer#1, Concern # 3: 

Author response: The overall organization of the manuscript is not discussed anywhere in the manuscript. Please add the same in the introduction section of the manuscript.  

Author action: We updated the manuscript by adding introduction of the overall organization.

Reviewer#1, Concern # 4: 

Author response: Introduction section must discuss the technical gaps associated with the current problem.  

Author action:We updated the manuscript by modify the introduction section . 

Reviewer#1, Concern # 5: 

Author response: The literature section is missing, authors are recommended to incorporate the same for better comprehensibility of the study.  

Author action: We updated the manuscript by modify the introduction. Add Reference 6-11 to illustrate the need for this research. 

Reviewer#1, Concern # 6: 

Author response: Authors may include some of the relevant studies on deep learning like https://doi.org/10.3390/bios12060393 and https://doi.org/10.3390/s22082988. 

Author action: Yes, we consulted these two articles and got great help. And we added them to the list of references.

Reviewer#1, Concern # 7: 

Author response: Authors may provide the architecture/block diagram of the proposed model for better comprehensibility of the proposed model concerning to various aspects of the proposed model.  

Author action: Yes, we modified this part.

Reviewer#1, Concern # 8: 

Author response: More explanation of the proposed model is desired on technical grounds.  

Author action: Yes, we modified this part.

Reviewer#1, Concern # 9: 

Author response: What is the size of the input image that is considered for processing and the size of the kernels?  

Author action: Yes, we modified this part.We updated the manuscript by update table 1. 

Reviewer#1, Concern # 10: 

Author response: The important details like the size of the input/tensor/kernel must be discussed, and whether authors have used Stride 1 or Stride 2 must be presented. What type of activation function is being used in the current study.  

Author action: Yes, we modified this part.We updated the manuscript by update table 1. 

Reviewer#1, Concern # 11: 

Author response: For how many epochs does the proposed model executed. what is the initial learning rate and after how many epochs does the model's learning rate saturated.  

Author action: Yes, we modified this part.We updated the manuscript by update table 2. 

Reviewer#1, Concern # 12: 

Author response: Authors may present the loss functions for better comprehensibility of each of the models used in the proposed model.  

Author action: Yes, we modified this part.

Reviewer#1, Concern # 13: 

Author response: Majority of the figures lack the clarity, they quality is fair but they must be explained in the text and the figures must be cited. Where is the graph for testing loss and accuracy presented in the study.  

Author action: Yes, we modified this part.

Reviewer#1, Concern # 14: 

Author response: Please discuss more on the implementation platform and the dataset details as two sub-sections in the manuscript.  

Author action: Yes, we modified this part.We updated the manuscript by update the details of the implementation platform and the dataset details.

Reviewer#1, Concern # 15: 

Author response: What are the cases assumed as TP, TN, FP, FN (confusion matrix) in the current study. 

Author action: Yes, we modified this part. You can find them on page 9 of the updated manuscript.

Reviewer#1, Concern # 16: 

Author response: Authors must provide the details of hyper parameters like training loss, testing loss, training accuracy and testing accuracy.  

Author action: Yes, we modified this part.We updated the manuscript by update table 3.

Reviewer#1, Concern # 17: 

Author response: More comparative analysis with state-of-art models is desired.  

Author action: Yes, we modified this part.You can find them on page 11 of the updated manuscript.

Reviewer#1, Concern # 18: 

Author response: By considering the current form of the conclusion section, it is hard to understand by PLOS ONE Journal readers. It should be extended with new sentences about the necessity and contributions of the study by considering the authors' opinions about the experimental results derived from some other well-known objective evaluation values if it is possible.  

Author action: Yes, we modified this part. We updated the manuscript by modify the conclusion section.

Reviewer#2, Concern # 1: 

Author response: Both the abstract and introduction are not professionally written. Some sentences in abstract should be modified to make it more attractive for readers.

Author action: Yes, We updated the manuscript by modify the abstract . 

Reviewer#2, Concern # 2: 

Author response: In Introduction section, it is difficult to understand the novelty of the presented research work. This section should be modified carefully. In addition, the main contribution should be presented in the form of bullets.

Author action: We updated the manuscript by the contribution. We have added the contribution of this study and other changes.

Reviewer#2, Concern # 3: 

Author response: The authors should also discuss the following works, which are about deep neural networks and image classification: ‘A Two-Tier Framework Based on GoogLeNet and YOLOv3 Models for Tumor Detection in MRI’, ‘Diabetic Retinopathy Detection Using VGG-NIN a Deep Learning Architecture’, ‘Traffic accident detection and condition analysis based on social networking data’, ‘Classification of skin disease using deep learning neural networks with MobileNet V2 and LSTM’

Author action: Yes, we consulted these two articles and got great help. And we added them to the list of references.

Reviewer#2, Concern # 4: 

Author response: Figures are blurred, it is difficult to read these figures, their quality should be improved (see fig 3)

Author action: Yes, we modified this part.We have updated Figure 3.

Reviewer#2, Concern # 5: 

Author response: Captions of the Figures not self-explanatory. The caption of figures should be self-explanatory, and clearly explaining the figure. Extend the description of the mentioned figures to make them self-explanatory.

Author action: Yes, We updated the manuscript by modifying some picture’s name.

Reviewer#2, Concern # 6: 

Author response: The conclusion section should be revised. In addition, the future work should be properly discussed.

Author action: Yes, we modified this part. We updated the manuscript by modify the conclusion section. 

Reviewer#2, Concern # 7: 

Author response: The whole manuscript should be thoroughly revised in order to improve its English.

Author action: I am very sorry for the trouble I have caused you.We have sent our manuscript to our  proofreader for revision.

---

## [Decision Letter · Decision Letter 1]

20 Mar 2023

Hazard source detection of longitudinal tearing of conveyor belt based on deep learning

PONE-D-22-30806R1

Dear Dr. Wang

We’re pleased to inform you that your manuscript has been judged scientifically suitable for publication and will be formally accepted for publication once it meets all outstanding technical requirements.

Kind regards,

Muhammad Fazal Ijaz

Academic Editor

PLOS ONE

Additional Editor Comments (optional):

Reviewers' comments:

Reviewer's Responses to Questions

**Comments to the Author**

1. If the authors have adequately addressed your comments raised in a previous round of review and you feel that this manuscript is now acceptable for publication, you may indicate that here to bypass the “Comments to the Author” section, enter your conflict of interest statement in the “Confidential to Editor” section, and submit your "Accept" recommendation.

Reviewer #1: All comments have been addressed

2. Is the manuscript technically sound, and do the data support the conclusions?

Reviewer #1: Yes

3. Has the statistical analysis been performed appropriately and rigorously? 

Reviewer #1: Yes

4. Have the authors made all data underlying the findings in their manuscript fully available?

Reviewer #1: Yes

5. Is the manuscript presented in an intelligible fashion and written in standard English?

Reviewer #1: Yes

6. Review Comments to the Author

Reviewer #1: The authors have addressed all the recommendations of the reviewers in a reasonable manner, manuscript in the current from may be considered for the further phase of editorial process.

7. PLOS authors have the option to publish the peer review history of their article (what does this mean?). If published, this will include your full peer review and any attached files.

Reviewer #1: No

---

## [Editor Report · Acceptance letter]

28 Mar 2023

PONE-D-22-30806R1 

Hazard source detection of longitudinal tearing of conveyor belt based on deep learning 

Dear Dr. Wang:

I'm pleased to inform you that your manuscript has been deemed suitable for publication in PLOS ONE. Congratulations! Your manuscript is now with our production department. 

Kind regards, 

on behalf of

Dr. Muhammad Fazal Ijaz 

Academic Editor

PLOS ONE